# Children’s Symptoms with a Febrile Illness and a Positive or Negative Test of SARS-CoV-2 during the Omicron Wave

**DOI:** 10.3390/children10030419

**Published:** 2023-02-22

**Authors:** Ricarda Möhler, Ekkehart Jenetzky, Silke Schwarz, Moritz Gwiasda, Larisa Rathjens, David D. Martin

**Affiliations:** 1Faculty of Health, School of Medicine, 58448 Witten, Germany; 2Department of Child and Adolescent Psychiatry and Psychotherapy, University Medical Center of the Johannes-Gutenberg-University Mainz, 55131 Mainz, Germany; 3Department of Pediatrics, Eberhard-Karls University Tübingen, 72076 Tübingen, Germany

**Keywords:** fever of unknown origin, children, SARS-CoV-2 variants, ecological momentary assessments, FeverApp

## Abstract

Febrile infections are common in childhood. Children can be infected with SARS-CoV-2, but their course is milder than in adults. So far, a comparison between febrile infections with a positive or negative Corona test with the Omicron variant is missing. The data used are from the FeverApp registry, which collects parent-reported data on febrile infections in children and informs about fever management. A comparison of symptomatic differences between episodes with a positive or negative Corona test was performed using Χ^2^-tests. During the Omicron wave, reported tests doubled and positive test results nearly 12-folded. In episodes with positive Corona saliva tests, more cough, fatigue, disturbed smell/taste, limb pain, sore throat, signs of serious sickness, and touch sensitivity were reported. Children with a negative Corona test show more tonsillitis, teething, any pain symptoms, earaches, and rashes. Thus, there are some significant differences between febrile infections with a positive or negative Corona test, but symptoms are present on both sides. The omicron variant seems to be more infectious than the alpha or delta variants in children, but the symptoms remain mild and do not differ much from other febrile infections.

## 1. Introduction

Febrile infections are common in young children [1,2] and respiratory infections are among the most common, especially when they attend daycare centers [3,4]. Children younger than 2 years may have as many as four to ten respiratory infections per year [3]. The number of febrile infections decreases with age [2]. COVID-19 is also a respiratory infection [5] that appears to be milder in children than in adults [4,5,6,7]. The most common symptoms in children with Corona are fever, coughing, vomiting, diarrhea, rhinitis, sore throat, myalgia, headache, fatigue, and a disturbed smell/taste, which was reported in several studies [4]. Around 20% of children are asymptotic [4].

Studies show that the different Corona variants can have different infectivity, severity, and symptomatology in adults [8,9] as well as in children [9,10,11]. At the onset of the omicron wave, hospitalization rates in children increased in some countries, leading to concern [11,12,13]. Data from Germany, England, and the U.S. show that the hospitalization rate of children did not increase [14,15,16,17], but the infection rate in Germany very sharply increased in early January 2022 [18]. In Denmark, it was found that especially in younger children (0–2 years), the infection rate increased [15]. The severity was lower than in the delta variant, but between age groups, younger children showed more severity [19]. The symptomatic manifestations of the Omicron variant in children differed from other variants in some studies [10,20], with more diagnoses in the upper respiratory tract than in the lungs [20]. Because the omicron variant has a better replication in the upper respiratory tract than in the lower respiratory tract [21,22], this could explain why it also spreads more rapidly even in children, and shows more upper respiratory infections than in the lungs [20].

Because the omicron variant is still the predominant variant, continued monitoring of this variant is important [23]. Therefore, it remains important to demonstrate the infection rate of Omicron and its specific symptomatic manifestations. A comparison between other febrile infections offers the possibility of a better distinction between Corona infections and other infections from a symptomatic point of view and shows the severity of symptomatology. Because febrile infections mainly occur in early childhood, specific consideration of different age groups would also be important.

Therefore, the aim of this study was the following:Compare the rate of Corona infections before and within the Omicron wave.Compare the symptoms of a febrile infection with a negative or positive Corona test during the Omicron wave.Compare the symptoms between positive and negative Corona tests within age groups during the Omicron wave.

Since 2019, an ecological momentary assessment (EMA) of febrile infections exists with the FeverApp registry, which contains documentation of Corona tests performed on participating families since April 2020 as well. A study has shown that it can extend the data collected at pediatric practices [24]. To shed light on the research questions raised above, the FeverApp data are used as it may complement studies conducted in other settings.

## 2. Materials and Methods

### 2.1. Data Collection

The FeverApp is an EMA of fever management in families. The app functions as a documentation and information tool where parents and other caregivers can document a child’s febrile episode and inform themselves about fever management in children.

The main objective is to assess guideline adherence regarding fever management in children [25]. The FeverApp launched in September 2019, and by the end of September 2022, more than 19,146 fever episodes from 8592 children were recorded. The collected data are stored on a server at the university and can be used for various scientific questions.

The family is the main object of observation, and caregivers can be distinguished into different documenting roles (mother, father, grandmother, grandfather, etc.). Observed children can be entered as different profiles if one has more than one child. Multiple time points within a fever episode as well as multiple fever episodes can be documented for each child.

Sociodemographic parameters such as the caregiver’s age, nationality, school degree, as well as children’s age and gender, are recorded during the initial entry. Children’s medical history such as the amount and duration of previous fever episodes as well as chronic diseases are also queried during the initial recording. Acute fever-related parameters during a fever episode, such as vaccinations or Corona infections, temperature, well-being, symptoms, warning signs, measures, and medication intake are continuously queried for each entry.

### 2.2. Study Sample

Only German citizens were selected, as the app is also used in other countries. Since there is no age restriction in the FeverApp and parents can report their children’s fever even when they are 18 years and older, these were filtered as well to provide an exclusive view of pediatric symptoms during a Corona infection.

For the first question, episodes of all children, episodes with a reported Corona saliva test and episodes with a positive result of the Corona saliva test between April 2020 and September 2022 were compared. It should be kept in mind that the number of children observed increases with the passage of time and, in some cases, the use of the FeverApp is also terminated. Furthermore, fever episodes are subject to the seasonal and pandemic measure-related variation that is not evident in the cumulative plot.

The main question is to compare symptoms between episodes with a reported positive or negative test during the Omicron wave. Fever episodes were chosen as the data level because the same child may differ in febrile infections with a negative or positive test. Therefore, only episodes with a report of a negative or positive test were analyzed, and episodes without a test or with a pending test were filtered out. The date was split between April 2020 to December 2021 and January 2022 to September 2022 to allow analysis of the omicron wave. For the main analyses with the Omicron wave, 3295 episodes were finally observed.

### 2.3. Variable Description

Various symptoms are questioned within the FeverApp. For this analysis, general symptoms, symptoms of pain, and symptoms that can be interpreted as warning signs as well as temperature as an indicator of fever were considered. Not only specific Corona symptoms were chosen [26], but all the symptoms that are queried in the app to get a broader overview of the symptoms. Multiple symptoms can be selected in different symptom areas. For this analysis, symptoms of an area were defined as prevalent if they were reported at least once in an episode. Not all of the evaluated symptoms are typical during a Corona infection [4], but all of them were selected for the analysis to better distinguish against other febrile infections and potentially expand the symptoms of the Omicron variant. Children’s well-being and parents’ confidence in managing their child’s fever were added to get an overview of the perceived severity of the illness.

Temperature can be entered as an exact number. For the analyses, the mean temperature within an episode and the maximum temperature within an episode were chosen. The variable “Fever” had to be operationalized using the maximum temperature, and the threshold of 38.5 °C was defined as fever. Questions regarding general symptoms included the following: “Does your child have symptoms?” with the answer options “no” or “cough”, “constrained breathing”, “freezing/chills”, “fatigue”, “sense of smell/taste disturbed”, “mucus”, “tonsillitis”, ”joint swelling”, “teething”, and “other symptoms”. Pain symptoms were determined by “Does your child have pain?” and the answer options “no” or “yes, in the limbs”, “yes, in the head”, “yes, in the neck”, “yes, in the ear”, “yes, in the stomach”, and “yes, somewhere else”. Symptoms that were defined as warning signs were questioned in several ways. On one hand, “Does <Name of the child> have (other) warning signs?” was asked, which can be answered with “no” or “yes, touch sensitivity”, “yes, shrill screaming like I have never heard it before”, “yes, acting differently, apathy”, and “yes, seems seriously sick”. Specific warning signs like diarrhea/vomiting were dichotomized as having “no diarrhea or vomiting” and “having diarrhea and/or vomiting”. Rashes were dichotomized as “no rash” and “yes, the redness can be pushed away or cannot be pushed away”. The parameter “dehydration” was another form of a warning sign, based on the question “Are there signs of dehydration?”, with the answer options of “no”, “yes, dry mucous membranes”, “yes, the skin is dry and loose”, “yes, child looks very tired”, ”yes, the eye-sockets appear sunken”, “yes, significantly less wet diapers than normal”, and “yes, the fontanelle is sunken”. The well-being of the child was questioned with a 5-point scale of sad to smile faces questioning “How does your child feel”? —Parents’ confidence was questioned with a 5-point scale of thumbs down to thumbs up questioning “When your child has a fever, do you feel confident that you can deal with it approriately?”. The minimum and maximum entries within an episode were compared.

### 2.4. Statistical Analyses

IBM SPSS, Version 28 (IBM, North Castle, NY, USA) was used for data analysis. For the first question regarding testing and infection rate before and with the Omicron variant, monthly reported episodes were compared with monthly reported testing of a Corona saliva test as well as monthly reported positive test results. For the comparison of symptoms during the Omicron wave, the dataset was split between April 2020 to December 2022 and January 2022 to September 2022 based on the RKI reports on the Omicron wave in Germany [27]. Three different analyses were performed regarding the symptoms during the Omicron wave. First, a survey of the symptoms (symptoms reported at least once within an episode) during the Omicron wave with a positive or negative Corona saliva test. Second, a symptomatic comparison between different age groups within the group with a positive Corona saliva test. Finally, a comparison of symptoms with a positive and negative Corona saliva test within different age groups.

For the symptomatic comparison, Χ^2^-Test and logistic regression were used and odds ratios were reported. When the sample was too small, the Fisher exact test was used. For metric variables, t-test and Mann-Whitney-U-tests were used. All significance levels were 5%.

## 3. Results

### 3.1. Corona Testing and Infection Rate within the FeverApp

From September 2019 until September 2022, 18,589 fever episodes were reported in the FeverApp, 9121 (47.6%) in the Omicron wave. Since April 2020, parents could report a Corona test. A total of 5112 episodes with a Corona saliva test were reported until September 2022, 3402 (66.5%) during the Omicron wave. A positive Corona test was reported in 727 fever episodes, 697 during the Omicron wave alone, and 30 Corona tests were reported from April 2020 to December 2021. Because reported episodes before and during the Omicron wave accounted for essentially the same percentage (52.40% vs. 47.60%), the rate of testing doubled from 33.5% to 66.5% episodes with a test. Of these tests, 1.8% were positive before the Omicron wave and 20.5% were positive during the Omicron wave. This suggests a nearly 12-fold increase in the number of positive tests. Figure 1 shows the monthly reported episodes with febrile infections, a Corona saliva test, and a positive Corona saliva test result in the FeverApp. An increase in positive test results can be seen around January 2022, although the monthly testing rate was already increasing around August 2021.

### 3.2. The Omicron Wave—Descriptive Statistics of the Children

From January 2022 to September 2022, 2432 children were recorded. Around one-quarter (27.00%) had a positive Corona test. For these children, 3295 episodes were recorded, and a positive Corona test was detected in 697 (21.20%) episodes. More children (53.20%) were male, but there was not a significant sex difference between the positive and negative test groups. The median age was 2 years, and there were significantly more children with positive Corona tests in the group of 0–2 years old compared with the group of 3–5 years old children. Chronic diseases were found in 4.00% of children, with no significant differences between the positive or negative Corona test groups. For further details of descriptive statistics, see Table 1. 

### 3.3. The Omicron Wave—Symptomatic Manifestation in Febrile Episodes with a Positive or Negative Corona Test

With a positive Corona saliva test, the most commonly reported symptoms during an episode were fever based on the maximum temperature (72.3%), fatigue (68.9%), cough (60.7%), and mucus (39.1%). Other symptoms included constrained breathing (24.1%), pain in the limbs (22.8%), headache (21.1%), and a sore throat (21.1%). Diarrhea was also reported in almost one-fifth of cases (19.2%). A further overview of the symptoms during the Omicron wave can be found in Appendix A.

The mean temperature within episodes was 38.6 °C with a positive test and 38.7 °C with a negative test, the difference was significant (t(3130) = 3.736; *p* < 0.001). Fever (threshold 38.5 °C) based on the maximum temperature within the episode was reported in 72.30% of episodes with a positive Corona test and in 70.10% of episodes with a negative Corona test. The difference was not significant.

In episodes with a positive Corona saliva test, more coughing (OR = 1.891, 95% CI: 1.583–2.259; *p* < 0.001), fatigue (OR = 1.411, 95% CI: 1.171–1.700; *p* < 0.001), and disturbed smell or taste (OR = 4.798, 95% CI: 2.498–9.214; *p* < 0.001) were reported (Figure 2). More signs of serious sickness (OR = 1.359, 95% CI: 1.038–1.780; *p* = 0.025) and touch sensitivity (OR = 1.846, 95% CI: 1.269–2.687; *p* < 0.001) were reported (Figure 3). Lastly, more pain in the limbs (OR = 1.622, 95% CI: 1.280–2.057; *p* < 0.001) and a sore throat (OR = 1.393, 95% CI: 1.094–1.773; *p* = 0.007) were reported as well (Figure 4).

Children with a negative Corona saliva test showed more tonsillitis (OR = 0.290, 95% CI: 0.105–0.804; *p* = 0.008), teething (OR = 0.689, 95% CI: 0.105–0.804; *p* = 0.004), any pain symptom (OR = 0.744, 95% CI: 0.607–0.911; *p* = 0.004), pain in the ears (OR = 0.401, 95% CI: 0.229–0.702; *p* < 0.001), and rash (OR = 0.463, 95% CI: 0.291–0.734; *p* < 0.001). There were no significant differences between positive and negative tests for signs of dehydration (Figure 5). The minimum well-being of children did not significantly differ between positive or negative tested children (*p* = 0.865). The maximum well-being of children was 3.12 in positive-tested children and 2.80 in negative-tested children, which is a significant difference (*p* < 0.001). Parents’ confidence, on the other hand, was significantly lower at minimum entries (*p* = 0.008) but significantly higher at maximum entries (*p* = 0.042) when children had a positive corona test. 

### 3.4. The Omicron Wave—Symptomatic Manifestation in Febrile Episodes with a Positive or Negative Corona Test in Different Age Groups

Regarding a positive Corona test, there were significant differences between age groups. Any warning sign (13.46%), shrill screaming (9.65%), less wet diapers (7.30%), cough (63.70%), constrained breathing (29.80%), and teething (21.24%) were most frequently reported in the group of 0–2-year-old children. Fever was most frequently reported in the group of 3–5-year-old children (77.84%). For the group of 6–17-year-old children, more diarrhea (28.75%), any pain symptom (55.00%), headache (62.50%), sore throat (33.75%), stomachache (25.00%) freezing/chills (43.75%), and other symptoms (16.30%) were reported. Further results can be seen in Appendix A.

There were also significant differences when comparing positive and negative test results within age groups. In the group of 0–2 years old children, significantly more children with a positive test result appeared seriously sick (*p* = 0.002), had pain in the limbs (*p* < 0.001), a sore throat (*p* = 0.002), had coughing (*p* < 0.001), fatigue (*p* = 0.003), and a disturbed smell or taste (*p* < 0.001). A rash (*p* = 0.003), pain in the ears (*p* = 0.025), pain somewhere else (*p* = 0.015), and teething (*p* < 0.001) were more frequently reported with a negative test result.

In the group of 3–5-year-old children, touch sensitivity (*p* = 0.005), signs of dehydration such as sunken eye sockets (*p* = 0.049), coughing (*p* = 0.005), and disturbed smell or taste (*p* = 0.018) were reported more with a positive test result. Any warning sign (*p* = 0.023), a rash (*p* = 0.011), and any pain symptom (*p* = 0.046) were more frequently reported with a negative test result.

In the group of 6–17-year-old children, diarrhea (*p* = 0.009), headache (*p* = 0.047), freezing/chills (*p* = 0.004), and a disturbed smell or taste (*p* = 0.022) were more frequently reported with a positive test. Tiredness (*p* = 0.028) was more often reported when the test was negative. Appendix A shows the significant differences between age groups with a positive or negative Corona test.

## 4. Discussion

The main objective of the study was to compare the symptoms of a febrile infection in children with a positive or a negative Corona test during the Omicron wave. Second, to compare the rate of infection during the Omicron wave with the waves before it and observe possible age differences during the Omicron wave. This analysis potentially shows the symptomatic manifestations and severity of Omicron and is able to distinguish the specific symptoms of Omicron compared with those of other febrile infections.

### 4.1. Corona Testing and Infection Rate within the FeverApp

The number of tests doubled in the Omicron wave. Positive results of a Corona test increased almost twelvefold during the Omicron wave. Although the number of tests doubled, this is not the only explanation for the high number of positive test results.

In silico analyses showed that viral infectivity of the Omicron variant increased 13-fold and that Omicron is about 2.8 times more infectious than the Delta variant [28]. A study from South Korea [29] showed that the susceptibility to the Omicron variant was 5-fold higher compared with the pre-delta variant and 3-fold higher compared to the delta variant. Another study from Germany [10] showed partially similar results to our findings. There were 23.6% positive tests in the Omicron wave and 2.5% positive test results before Omicron. In general, the rate of positive tests was 4.8%, whereas in our study it was 14.2%. This is explained by the tenfold higher amount of testing before the Omicron wave in the cited study.

In our study, children aged 0–2 years were the largest group with a positive Corona test (56.4%). Other studies show different results. A study from Spain showed that most children with a positive test were 5–12 years old, and the second largest group was 0–4 years old (43.1%) in the Omicron wave [20]. In a study from Germany, the mean age drops from 10.7 years to 8.0 years from the delta to the Omicron wave, which is also the youngest age in all waves [10]. Nevertheless, at the beginning of the Omicron wave in Denmark, the incidence of SARS-CoV-2 cases was also highest in the 0–2-year-old group [30].

One explanation could be that not only the study sample but also the overall FeverApp sample is generally very young. The FeverApp is mainly used for febrile infections, which are more common in younger children [1,2,3]. It is also distributed through pediatric practices. Many of them recommend the FeverApp during the U-Examinations, which are performed from a child’s first days to 5 years of age [31].

### 4.2. The Omicron Wave—Symptomatic Manifestation in Febrile Episodes with a Positive or Negative Corona Test

Regarding the Omicron wave, the main symptoms are fever, fatigue, cough, mucus, constrained breathing, pain in the limbs, headache, and a sore throat as well as diarrhea. Compared with another study from Germany [10], rhinitis, cough, fever, a sore throat, and headache were the most reported symptoms, but fever and cough were 20–50% less frequently reported than in our study. In a study from France, similar numbers of symptoms such as fever, fatigue, and cough [11] were mentioned, and in a study from Spain, similar results were found regarding the frequency of fever, cough sore throat, and diarrhea [20]. The difference can be explained by the fact that our study did not ask questions about symptoms related to the nasal system. Second, we observed an entire fever episode and included the symptoms, which were recorded at least once within that fever episode. Third, the FeverApp showed to have a higher documentation and precision rate than records from a pediatrician’s practice [24]. Finally, the frequencies of Corona symptoms differ in other studies as well. A collection of different studies shows that the fever frequency varies from 36% to 73% and cough frequencies vary from 18% to 54% [32]. While adults seem to have fewer intestinal symptoms, they were more frequently observed in children [32]. In our study diarrhea (vomiting and/or diarrhea) occurred in 19.2% of cases, contributing to this observation.

Comparing Omicron symptoms with those of other febrile infections, episodes with a positive Corona saliva test show more cough, fatigue, disturbed smell or taste, pain in the limbs, a sore throat, signs of serious sickness, and touch sensitivity. Children with a negative Corona saliva test on the other hand show more tonsillitis, teething, any pain symptoms, pain in the ears, and rash.

The highest odds ratio was found in disturbed smell/taste with a four times higher probability of having disturbed smell or taste with a positive corona test. On the other hand, the frequency was very low from 0.7 to 3.2%. This shows that this is either a specific corona symptom, or parents might expect smell or taste disturbance more in connection with a corona infection, therefore it was reported more often.

The largest percentage difference was found in coughing, which was reported about 15.7% more often when children had a positive Corona test. It has to be added that respiratory infections are the most frequent infections in childhood [3,4], and therefore the symptoms recorded were most likely also from respiratory infections. However, there is a possibility that the children had different infections and therefore the differences were not due to the specificity of the Omicron infection, but rather to an infection in a different system. However, there are symptomatic differences on both sides and the percentage differences are not exceptionally high, so in this registry, children infected with Omicron are not necessarily sicker than children with other infections. The perceived maximum well-being in children with a positive corona test was also significantly higher than the maximum well-being in children with a negative test. Children’s minimum well-being did not significantly differ, which speaks for no difference in severity. Surprisingly, parents’ minimum confidence was lower in parents with a positive tested child, even though well-being was better or there was no difference. This might speak for an effect of a positive corona test on perceived confidence.

### 4.3. The Omicron Wave—Symptomatic Manifestation in Febrile Episodes with a Positive or Negative Corona Test in Different Age Groups

The most commonly reported symptoms during the Omicron wave differed in some cases between age groups. Coughing and constrained breathing were primarily reported in the group of 0 to 2-year-old children. Fever was reported primarily in the group of 3–5-year-old children. Diarrhea, headache, a sore throat, and stomachache were primarily reported in the oldest group, 6 to 17-year-old children. The extent of fatigue, pain in the limbs, and mucus did not significantly differ between age groups. An explanation for the lower reporting of pain symptoms in the group of 0–2-year-old children could be a communication problem since children may not be able to verbally communicate their pain.

When comparing symptoms between positive and negative Corona tests within age groups, the group of 0 to 2-year-old children showed the greatest differences between a positive or negative Corona saliva test. The differences decreased by age. In this study, it appears that younger children differ more in the symptomatology of febrile infections with the Omicron variant than with other febrile infections. In contrast, a study from England showed that most specific Corona symptoms were reported less frequently in young children, with the exception of fever, which was more frequently reported in young children than in adolescents or young adults [9].

### 4.4. Strengths and Limitations

The strength of this study is the ecological momentary assessment. Parents are able to continuously report children’s symptoms in the home environment. This allows us to look at an entire episode and see which symptoms were reported at least once within that episode. Second, the FeverApp had a higher documentation and precision rate than data from a pediatric practice [24], so the study complements the results from other study settings. The high number of symptoms queried is another strength. In other studies, only a few symptoms or specific Corona symptoms were queried that were already known. With our study, we had the potential to expand the symptomatic manifestations. The test rate can be seen as another strength since parents usually document febrile episodes when symptoms are prevalent, compared with mass use of corona testing without prevalent symptoms, as this increases the false positive test results.

We analyzed episodes, not infected children, and therefore the comparison with other studies might be difficult. A limitation could be that the chosen observation unit was episodes instead of children, as a child with several episodes might have been included several times in the analyses. On the other hand, we thought it would be interesting to examine whether the same child differs in different infections. Second, children were tested with a Corona saliva test, which is less valid than a PCR test. Parental perception also plays a role. Children’s symptoms are only recorded by caregivers, who may be biased by the positive Corona test result and overinterpret specific symptoms that they believe are specific for a Corona infection. Finally, the very young age distribution of the children questions the generalizability and comparability of the study results. The vaccine status of the children was not part of the analyses because we only recorded recent vaccinations in relation to a new fever episode. In addition, these vaccinations were authorized for children from the age of 5 years onward in November 2021 [33] and for children from the age of 6 months onward in October 2022 [34]. Due to the fact that fever mainly occurs in the age below five years, presumably, almost all have been unvaccinated against SARS-Cov-2 at the time of the sample collection.

## 5. Conclusions

The results of our study show an almost 12-fold increase in infection in children during the Omicron wave compared with the waves before. Age differences were noted in infection rate and symptomatic manifestations but should be viewed with caution in terms of generalizability because of the specific febrile and symptomatic very young sample. A comparison between fever episodes with a positive or negative Corona test shows significant differences in some symptomatic manifestations, arguing for a differentiation between febrile infections with a Corona infection or another febrile infection. The percentage differences were not exceptionally large, higher frequencies of symptoms were seen on both sides, and children’s well-being did not differ between the groups or was even higher in positive-tested children. This argues for no specific symptomatic severity of an Omicron-Corona infection compared with other febrile infections.

## Figures and Tables

**Figure 1 children-10-00419-f001:**
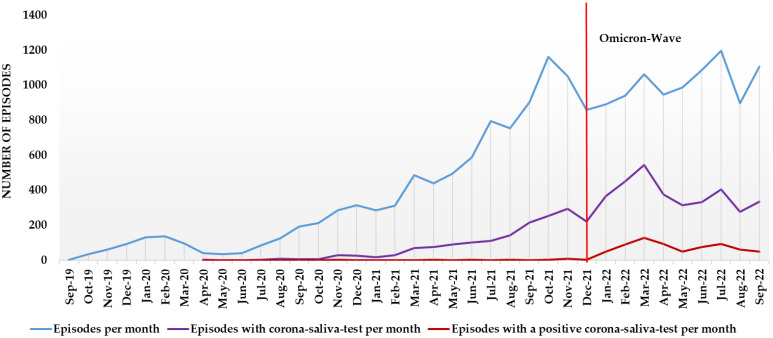
Monthly number of febrile episodes, Corona tests, and positive Corona test results within the FeverApp.

**Figure 2 children-10-00419-f002:**
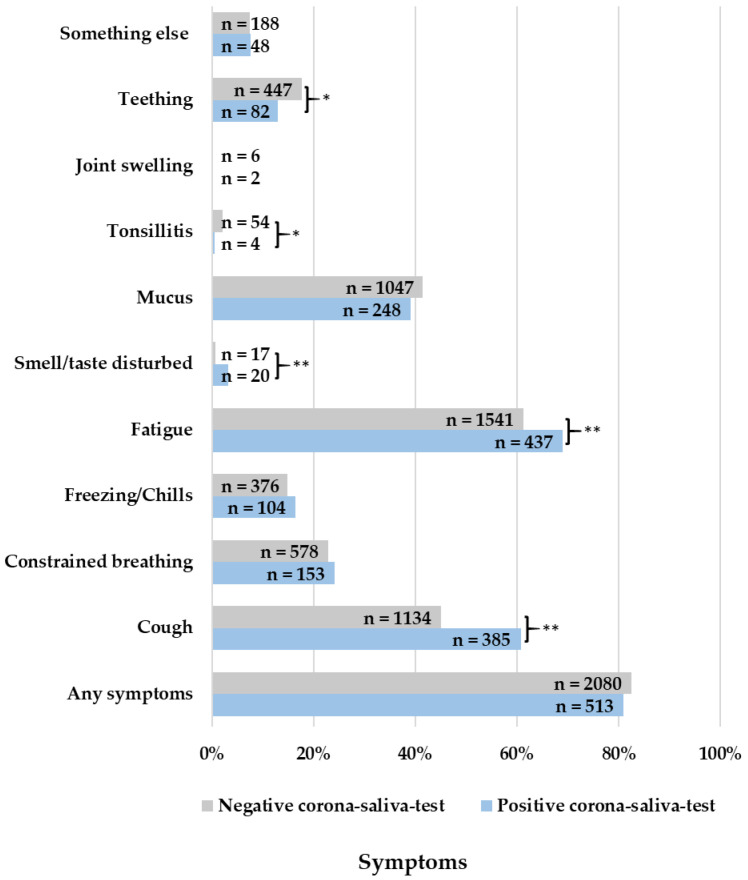
Reported symptoms in fever episodes, * = *p* < 0.05; ** = *p* < 0.001.

**Figure 3 children-10-00419-f003:**
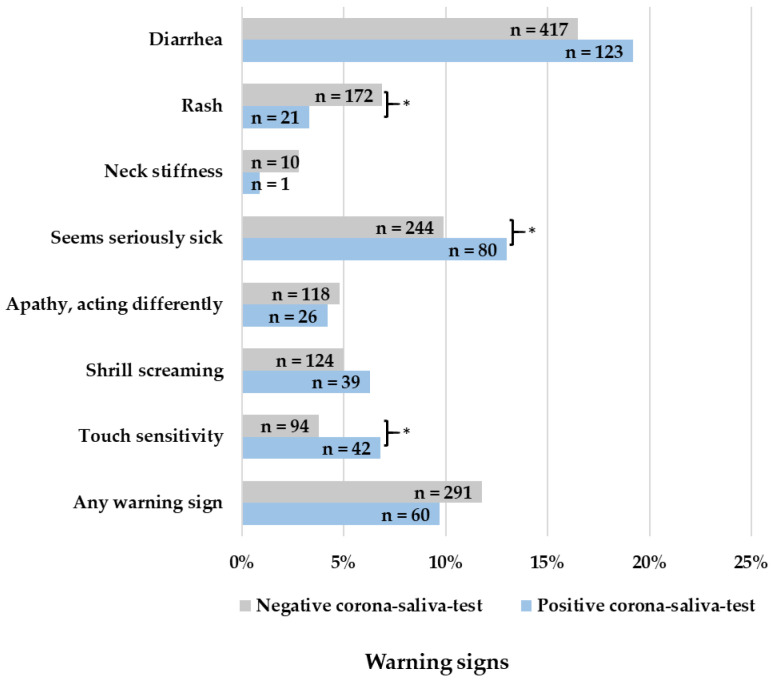
Reported warning signs in fever episodes, * = *p* < 0.05.

**Figure 4 children-10-00419-f004:**
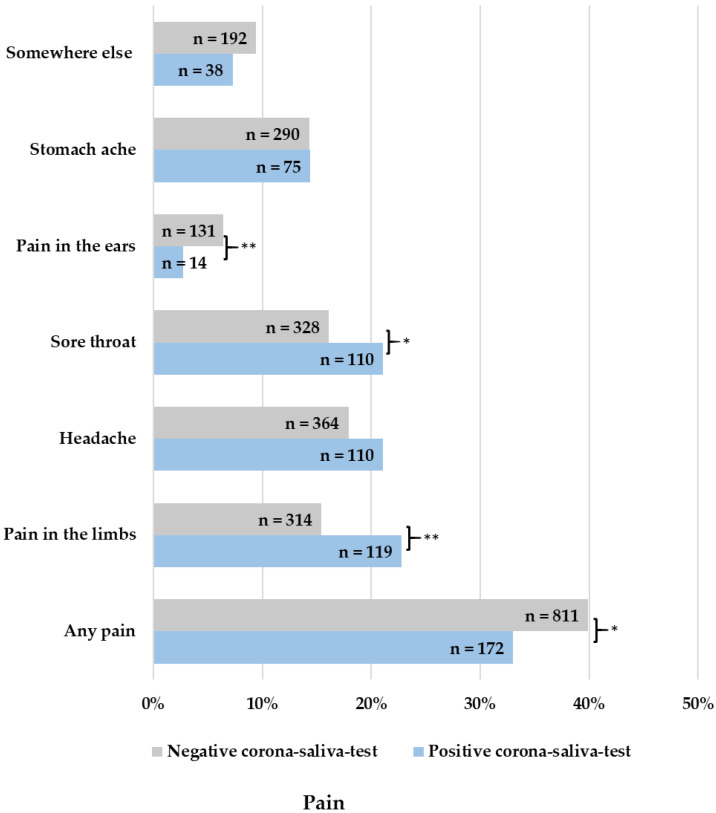
Reported pain in fever episodes, * = *p* < 0.05; ** = *p* < 0.001.

**Figure 5 children-10-00419-f005:**
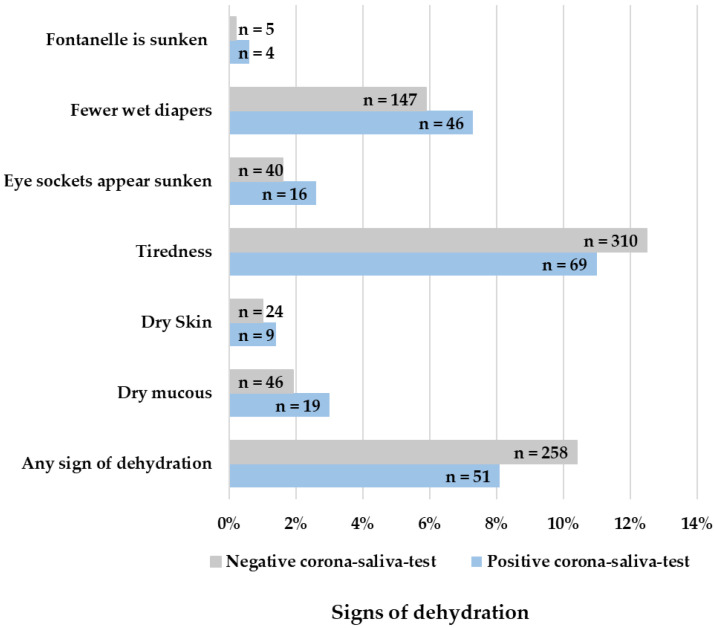
Reported signs of dehydration in fever episodes.

**Table 1 children-10-00419-t001:** Descriptive statistics of children with a positive or negative Corona saliva test during the omicron wave.

	Total	Negative Test	Positive Test	*p*-Value *	Odds-Ratio
Number of children	2432	1776 (73.00%)	656 (27.00%)		
Fever episodes	3295	2598 (78.80%)	697 (21.20%)		
Sex					
Male	1293 (53.20%)	951 (53.60%)	342 (52.10%)		
Female	1137 (46.80%)	823 (46.40%)	314 (47.90%)	0.518	1.061 [0.887–1.269]
Age in categories					
0–2 Years	1216	846 (69.60%)	370 (56.40%)		
3–5 Years	901	704 (78.10%)	197 (21.90%)	<0.001	0.640 [0.524–0.781]
6–17 Years	315	226 (71.70%)	89 (28.30%)	0.453	0.900 [0.685–1.184]
Chronical Diseases	95 (4.00%)	75 (4.30%)	20 (3.10%)	0.189	0.714 [0.432–1.180]

Note: * = *p*-Value based on bivariate logistic regression, categorial covariates have the first category as the reference.

## Data Availability

The datasets analyzed during the current study are available from the corresponding author upon reasonable request.

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
