# Peer review of "Children’s Symptoms with a Febrile Illness and a Positive or Negative Test of SARS-CoV-2 during the Omicron Wave"

_children, 2023, doi:10.3390/children10030419_

Round 1
Reviewer 1 Report
Dear author,
The article is a well written one. It needs some modification:
1.The percentage differences were not exceptionally large, and higher frequencies of symptoms were seen on both sides, which does not argue for a specific symptomatic severity of a Corona infection compared to other febrile infections. The authors should include also a chart with the diagnostic of the patients.
Author Response
Dear author,
The article is a well written one. It needs some modification:
1.The percentage differences were not exceptionally large, and higher frequencies of symptoms were seen on both sides, which does not argue for a specific symptomatic severity of a Corona infection compared to other febrile infections. The authors should include also a chart with the diagnostic of the patients.
Answer: Thank you for your advice, but only parent’s document the fever symptoms of their children in the app and have no specific diagnostic. Often the parents do not contact the physician of the child of a mild fever episode; therefore, we have no medical record of children’s diagnoses. Hence our chart are pure parent based symptoms.
Reviewer 2 Report
Dear authors, you realized a very comprehensive analysis, with appreciable efforts, of many cases on childrens with a great variability of symptoms.
Only one conclusion may be deduced from this study: symptomatic manifestations are more severe in omicron positive patients, than in corona negative !
But many explantions may be given for a such large variability of symptomatology, resulted from individual reaction to different viral infections. Virulence of different strains is a main factor: rhino-, adeno-, common cold corona, paramyxo !
Author Response
Dear authors, you realized a very comprehensive analysis, with appreciable efforts, of many cases on childrens with a great variability of symptoms.
Only one conclusion may be deduced from this study: symptomatic manifestations are more severe in omicron positive patients, than in corona negative !
But many explantions may be given for a such large variability of symptomatology, resulted from individual reaction to different viral infections. Virulence of different strains is a main factor: rhino-, adeno-, common cold corona, paramyxo !
Answer: Thank you very much for your suggestion, which has led us to add an interesting result to our manuscript. It cannot be concluded that the omicron is more severe, because we found the following interesting result: The maximum well-being of a child is higher (3.12 vs. 2.80 on a scale from 1 to 5) and the minimum confidence of the parents is lower (mean 3.54 vs. 3.67 on a scale from 1 to 5) when the child is positively tested for corona. This surprising result is probably parametric with T-test as well as non-parametric with U-test. We added this into the manuscript in the method (line 124-125 and 146-150), results (line 222-227) discussion (line 334-340) and conclusion (line 386-390) section.
Reviewer 3 Report
The work of Ricarda Möhler and colleagues is devoted to an important topic, the study of the relationship between febrile illness and the result of a test for covid. This is important because the virus changes rapidly, and if the first waves of COVID-19 were accompanied by a low incidence of children, then the Omicron wave was associated with a significant number of cases among children.
The objectives of the study were achieved. The work is well illustrated. The conclusions correspond to the obtained results.
Author Response
Thank you very much for your suggestion.
Reviewer 4 Report
First, congratulations on the development of the manuscript.
Basically, the study evaluated the effects of the wave caused by the omicron variant on infection rates and clinical manifestations observed in children. In addition, it compared these manifestations between children who had or did not have a positive saliva test for the virus.
The results presented are in accordance with the proposed objectives, as well as their conclusions. I only make a few suggestions/corrections noted:
1- In topic 2.2 of the methodology, the authors report that individuals aged 18 or over are considered children. Are 18-year-olds considered children? 18 years and older would be considered adults.
2- In Table 1, in the results, the percentages (in parentheses) of the age category, in the positive test column, need to be corrected.
3- I could not observe any comments about the vaccination status of these children. In the investigated period, were there groups that had already received an anti-COVID vaccine? At the beginning of the year 2022, was the vaccine available for children?
Author Response
First, congratulations on the development of the manuscript.
Basically, the study evaluated the effects of the wave caused by the omicron variant on infection rates and clinical manifestations observed in children. In addition, it compared these manifestations between children who had or did not have a positive saliva test for the virus.
The results presented are in accordance with the proposed objectives, as well as their conclusions. I only make a few suggestions/corrections noted:
1- In topic 2.2 of the methodology, the authors report that individuals aged 18 or over are considered children. Are 18-year-olds considered children? 18 years and older would be considered adults.
Answer: Thank you for the notice. You are right they are adults. We changed the wording since it is confusing. We worded it this way since the FeverApp is used by parents to report fever of their children and it can be used for children, in the view of their parents, which are older than 18 years (for example when they are disabled and live at home). I hope it is clearer now, see line 95-98.
2- In Table 1, in the results, the percentages (in parentheses) of the age category, in the positive test column, need to be corrected.
Answer: Thank you for your notice. We have corrected it.
3- I could not observe any comments about the vaccination status of these children. In the investigated period, were there groups that had already received an anti-COVID vaccine? At the beginning of the year 2022, was the vaccine available for children?
Answer: We cannot comment on COVID-Vaccination status because we only record recent vaccinations (in the last 2 weeks) at a new fever episode. If the vaccine was in the past 3 weeks before the fever episode it is not recorded anymore since we are more interested in whether the fever is a reaction of the vaccine and not the general vaccine status. In addition, these vaccinations were authorized in children from 5 years age in November 2021 and in children from 6 months age in October 2022. Due to the fact, that fever occurs in children mainly under 5 years of age, presumably almost all have been unvaccinated against SARS-Cov2 at the time of sample collection. Therefore, the vaccine status was no specific topic within the analyses. We added a short explanation into the discussion section (line 380-387).
Reviewer 5 Report
Thank you for your effort in conducting research.
The study assessed Children’s Symptoms with a Febrile Illness before and during the Omicron-Wave with a Positive or Negative Test of SARS-CoV-2.
The manuscript is well-written and has an excellent flow, and the tables and figures are informative. I have no recommendation.
Author Response

(The authors gave the same response as above.)
